# Hepcidin Reduction during Testosterone Therapy in Men with Type 2 Diabetes: A Randomized, Double-Blinded, Placebo-Controlled Study

**DOI:** 10.3390/biomedicines11123184

**Published:** 2023-11-29

**Authors:** Line Velling Magnussen, Louise Helskov Jørgensen, Dorte Glintborg, Marianne Skovsager Andersen

**Affiliations:** 1Department of Endocrinology and Metabolism, Odense University Hospital, 5000 Odense, Denmark; dorte.glintborg@rsyd.dk (D.G.); marianne.andersen1@rsyd.dk (M.S.A.); 2Department of Clinical Biochemistry and Pharmacology, Odense University Hospital, 5000 Odense, Denmark; louise.helskov.joergensen@rsyd.dk

**Keywords:** testosterone therapy, cardiovascular disease markers, type 2 diabetes mellitus

## Abstract

High hepcidin is linked to low-grade inflammation and lower iron levels. The consequences of testosterone replacement therapy (TRT) on inflammation and the risk of cardiovascular disease (CVD) are undetermined. We investigate the effect of TRT on the inflammatory cardiovascular risk markers hepcidin-iron, fibroblast growth factor 23 (FGF23)-phosphate-klotho, and calprotectin pathways. Methods: A randomized, placebo-controlled, double-blinded study at an academic tertiary-care medical center. Interventions were testosterone gel (TRT, *n* = 20) or placebo gel (*n* = 19) for 24 weeks. We included 39 men (50–70 years) with type 2 diabetes (T2D) on metformin monotherapy with bioavailable testosterone levels <7.3 nmol/L. Body composition was assessed with DXA- and MRI-scans; the main study outcomes were serum hepcidin-iron, FGF23, phosphate, klotho, and calprotectin. Results: Hepcidin levels decreased during TRT (β = −9.5 ng/mL, *p* < 0.001), lean body mass (β = 1.9 kg, *p* = 0.001) increased, and total fat mass (β = −1.3 kg, *p* = 0.009) decreased compared to placebo. Delta hepcidin was not associated with changes in lean body mass or fat mass. Iron and the pathways of FGF23-phosphate-klotho and calprotectin were unchanged during TRT. Conclusions: During TRT, the reduction in hepcidin was not associated with circulating iron levels, lean body mass, or fat mass; these findings suggested a direct anti-inflammatory effect of TRT and no indirect effect mediated through these factors.

## 1. Introduction

Type 2 diabetes mellitus (T2D) is associated with a high risk of cardiovascular disease (CVD) [1]. Chronic low-grade inflammation is a hallmark of T2D and inflammation contributes to the development of CVD [2]. Inflammation [3], central obesity [4], and T2D are associated with lower testosterone levels, and 58% of men with T2D have hypogonadism [5,6]. High endogenous testosterone in men is associated with a lower risk of CVD [7], but the effect of testosterone replacement therapy (TRT) on CVD, in patients at risk, is debated. TRT increases lean body mass and reduces fat mass [8,9,10], including abdominal subcutaneous fat [11,12], whereas no change is seen in insulin resistance, visceral fat, or hepatic fat [10,12,13]. HDL cholesterol and adiponectin levels decrease during TRT compared to placebo [10], which could suggest increased cardiovascular risk. A large randomized controlled trial (RCT) observed an increased number of cardiovascular events during TRT compared to placebo in old men with mobility limitations [14]. Therefore, since 2015, a warning regarding risk of CVD has been applied to testosterone products by the U.S. Food and Drug Administration advisory committee [15]. However, a recent meta-analysis reported no evidence of increased CVD risk during TRT [16] and additional data are needed to ensure the safety of TRT in T2D. We and others have previously reported decreased leptin and leptin/adiponectin ratio in men with T2D during TRT [12,17,18,19]. Decreased leptin may be of benefit regarding the risk of CVD [12], whereas decreased adiponectin levels might add to CVD risk [12,18].

Hepcidin is a peptide hormone produced in the liver [20]. Acute or chronic inflammation increases hepcidin as part of the acute phase reactant response [21]. Hepcidin is responsible for iron homeostasis and high hepcidin levels decrease circulating iron levels [21]. High hepcidin levels may predispose to atherosclerotic lesions and CVD by enhancing iron retention in vascular plaque macrophages, thus promoting foam cell formation and plaque instability [22]. Higher hepcidin levels were seen in 166 men with T2D on metformin mono-therapy compared to 146 healthy controls in an observation study [23]. Three previous placebo-controlled RCT studies showed decreased hepcidin after 1–3 months of TRT [17,24,25]; however, one of the RCTs reported unchanged hepcidin after 6 months of TRT compared to placebo [25] and another study only reported results of hepcidin after 3 months of TRT despite a study duration of 12 months [24].

The activation of the inflammatory pathways fibroblast growth factor 23 (FGF23)-phosphate-klotho [26,27] and calprotectin [28] could decrease circulating levels of iron, thus stimulating hepcidin production. We have previously reported that phosphate and calprotectin decreased during TRT in aging men with low bioavailable testosterone levels and without T2D [29]. The effect of TRT on the FGF23-phosphate-klotho or calprotectin pathways has not been investigated in T2D.

In the current study, we investigated changes in the inflammatory cardiovascular risk markers hepcidin-iron, FGF23-phosphate-klotho, and calprotectin pathways during 24 weeks of TRT in men with T2D; secondarily, we investigated whether possible changes in inflammatory markers during TRT could be a direct or indirect effect.

## 2. Materials and Methods

This 24-week, randomized, double-blinded, placebo-controlled trial was conducted at Odense University Hospital (Denmark) from April 2012 to November 2013. Men aged 50–70 years with T2D on metformin monotherapy for a minimum of three months and bioavailable testosterone levels <7.3 nmol/L were included. The study was approved by the local Ethics Committee (identifier: S-20120002) and the Danish Health and Medicines Authority (identifier: 2011-002102-73). The trial was declared in ClinicalTrials.gov (identifier: NCT01560546) and all patients gave written informed consent at the screening visit. The study population is reported in detail elsewhere [10].

These data represent a secondary analysis of our study that evaluated the effect of TRT on lean body mass in men with T2D and low bioavailable testosterone [10].

### 2.1. Study Design

Patients were randomly assigned to 5 g gel daily containing 50 mg testosterone, Testim^®^ (TRT, *n* = 22), or placebo (*n* = 21). Patients were increased to 10 g gel daily if bioavailable testosterone levels were <7.3 nmol/L after three weeks of treatment [10]. The patients were examined before and after 24 weeks of TRT. The sample size of the study was determined according to the anticipated effect of TRT on total lean body mass [9] with an assumption of type 1 error (α) = 0.05, type 2 error (β) = 0.1, SD = 1.3 kg, along with a 25% dropout rate, resulting in 20 patients in each group. The patients had fasting blood samples performed. In the present study, the primary outcome measures included changes in hepcidin-iron, FGF23, phosphate, klotho, and calprotectin. Two non-testosterone-related serious adverse events occurred in the study. Safety monitoring was handled externally to ensure continued blinding.

### 2.2. Biochemical Variables

Bioactive hepcidin 25 (EIA-5782) FGF23 (EIA-6060), soluble alpha-klotho (EIA-5605), and calprotectin (EIA-5111) were measured using enzyme-linked immunosorbent assays (ELISA) according to the manufacturer’s instructions. All kits were from DRG Instruments GmbH, Marburg, Germany. Hepcidin and FGF23 showed intra-assay CVs below 8% on kit-controls, and for hepcidin a CV% of 9.1 on the serum pool control material. Klotho showed a CV% of 10.4 on serum pool control, and no kit controls were included with the kit. Calprotectin had a CV between 18.7 and 21.7% on kit-controls and 7.1% on the serum-pool control material.

Phosphate and iron were analyzed on Cobas8000 (Cobas^®^, Roche, Basel, Switzerland) according to routine procedures, accredited under ISO 15189, using photometric assays (PHOS2 ver.2 #05171377 190 and IRON Gen.2 #05169291 190). The procedures included internal and external quality controls.

### 2.3. Lean Body Mass and Total Fat Mass Evaluated with Dual-Energy X-ray Absorptiometry (DXA)

Results for lean body mass and total fat mass were obtained with DXA scans using a Hologic Discovery device (GE & Hologic, Waltham, MA, USA), as described in detail and published before [10].

### 2.4. Regional Body Fat Mass Evaluated with Magnetic Resonance Imaging (MRI)

An MRI was performed with a 3.0 T high-field MR Unit (Phillips Achieva, Phillips Healthcare). Three abdominal slices and one femoral slice were achieved using an axial, T1-weighted gradient-echo sequence. In-house developed software using MATLAB (MathWorks, Natick, MA, USA) was applied for the automatic segmentation of the images, yielding subcutaneous abdominal adipose tissue (SAT, %fat of total abdominal volume), visceral adipose tissue (VAT, %fat of total abdominal volume), and thigh subcutaneous fat area (TFA, %fat of total thigh volume), as described in detail and published previously [12].

### 2.5. Hepatic Fat Content Evaluated with Magnetic Resonance Spectroscopy (MRS)

Single-voxel liver 1H MRS was performed to measure the hepatic fat content. MRS measurements were performed using a Philips Achieva 3.0 T MR scanner (Philips Healthcare). The MRS data were acquired using a SENSE XL torso coil with 16 channels, following shimming, with volumes of interest (30 × 30 × 30 mm^3^) manually placed within the right lobe of the liver (segment six or seven). The point-resolved spectroscopy (PRESS) technique was performed without water suppression (repetition time ms/echo time ms, 2000/35). We collected spectra during a single breath hold (17.5 s). The water peak and the major fat peak of methylene, located at 4.7 and 1.3 ppm, respectively, were automatically fitted by using a spectroscopic analysis package included in the Philips workstation, as described in detail and published previously [12]. Area ratios (hepatic fat/water ratio) were calculated for each patient. An experienced MR spectroscopist, who was blinded to the treatment allocation, reviewed automated spectral results.

### 2.6. Whole-Body Insulin Sensitivity Estimated with a Euglycemic–Hyperinsulinemic Clamp

After an overnight fast, a 2 h basal tracer equilibration period was followed by a 4 h period with insulin infusion at a rate of 40 U/m^2^/min. A [3-3H]-glucose infusion was used throughout the 6 h study, and [3-3H]-glucose was added to the glucose infusates to maintain plasma-specific activity constantly at baseline levels during the 4 h clamp period. By varying the infusion of 20% glucose based on bedside plasma glucose measurements every 10–20 min, plasma-glucose was kept constant at approximately 5.5 mmol/L. Steele’s non-steady-state formulas were used to calculate the rates of total glucose appearance and glucose disposal (Rd). Insulin-stimulated Rd was taken as an estimate of whole-body insulin sensitivity, as described in detail and published previously [10].

### 2.7. Statistical Methods

We performed per-protocol analyses. Descriptive statistics were performed, providing results expressed as arithmetic mean ± standard deviation, geometric mean (95% CI), or median (interquartile range) as appropriate for each group (TRT or placebo) at baseline and after 24 weeks of treatment. Differences in baseline values were analyzed using an unpaired t-test on normally distributed data. Wilcoxon rank-sum tests were conducted at baseline if data could not be transformed to normally distributed data using a natural logarithm. Outcome measurements were assessed with multiple linear regression analyses controlled for baseline values on normally distributed data for the placebo-controlled mean effect of intervention between groups (β). The models were checked with residual plots and Box–Cox analysis. Absolute changes during 24 weeks from baseline are given as delta values (∆). For nonparametric data, outcome measurements were assessed with delta values using the Wilcoxon–Mann–Whitney test for the comparison of treatment groups.

The tests were two-sided and results of *p* < 0.05 were considered statistically significant. Pearson’s bivariate correlation test or a non-parametric Spearman rank correlation were performed as appropriate for the analysis of correlations between the measured parameters (hepcidin-iron, FGF23, phosphate, klotho, and calprotectin) as well as correlations with previously obtained results for lean body mass, total fat mass, insulin sensitivity during clamp, hepatic fat, SAT, VAT, TFA, HDL-cholesterol, adiponectin, leptin, bioavailable testosterone levels, 17β-Estradiol, dihydrotestosterone, hemoglobin, and hematocrit [10]. Statistical analyses were performed with STATA, version 16.

## 3. Results

At baseline, TRT and placebo groups were comparable regarding all study outcomes (Table 1). In all, 21/39 patients were obese with a BMI of 30–39.9 kg/m^2^, whereas 15/39 had a BMI 25–29.9 kg/m^2^ and 4/39 patients had normal weight with a BMI < 25 kg/m^2^. In all, 10/20 had a BMI < 30 kg/m^2^ in the testosterone group and 8/19 had a BMI < 30 kg/m^2^ in the placebo group.

During TRT, hepcidin levels decreased compared to placebo after 24 weeks of treatment (Table 1). Iron, FGF23, phosphate, klotho, and calprotectin were unchanged during 24 weeks of TRT compared to placebo (Table 1).

Lean body mass, hemoglobin, and hematocrit increased, whereas total fat mass, HDL-cholesterol, adiponectin, and leptin decreased in the TRT group (Table 1), as reported previously [10,12]. Furthermore, insulin sensitivity and ectopic fat, viscerally and in the liver, were unchanged during TRT (Table 1), as described previously [10,12].

### Correlations

In the TRT-group, no significant correlations were found between delta values of hepcidin and iron, FGF23-phosphate-klotho, calprotectin, hemoglobin, hematocrit, HDL-cholesterol, adiponectin, leptin, insulin sensitivity, or body composition. In the TRT-group, there were no significant correlations between the changes in hormone levels (testosterone, 17β-estradiol, or dihydrotestosterone) and changes in hepcidin and iron, FGF23-phosphate-klotho, or calprotectin.

Positive correlations were observed between changes in testosterone levels (total testosterone, bioavailable testosterone, and free testosterone) and 17β-estradiol (r = 0.64, *p* = 0.0025; r = 0.67, *p* = 0.0011; r = 0.66, *p* = 0.0014) and changes in dihydrotestosterone (r = 0.82, *p* < 0.001; r = 0.86, *p* < 0.001; r = 0.85, *p* < 0.001) in the testosterone group, as previously reported [10,12].

## 4. Discussion

The main finding of the present study was a significant reduction in hepcidin following 24 weeks of TRT compared to placebo in men with T2D and low bioavailable testosterone levels. The decrease in hepcidin was not related to changes in circulating iron levels or body composition; our results could suggest a direct effect of TRT on the synthesis of hepcidin from the liver, but this is only speculative. Our hepcidin levels were comparable to those in healthy men [23], which may indicate less-prevalent inflammation in our cohort.

Our finding of reduced hepcidin during TRT for 24 weeks added significant knowledge to previous RCTs on the topic of aging men with T2D [17], mobility problems [25], and anemia [24]. Hepcidin levels decreased during 1–3 months TRT in all three studies [17,24,25], but was unchanged compared to placebo after 6 months in the study by Bachman et al. [25] and was not reported in the study by Artz et al. [24]. Interestingly, the included study populations differed significantly in previous studies [17,24,25]. Our present cohort included men (aged 50─70 years) with T2D treated with continuous, stable metformin monotherapy throughout the study, which implied well-regulated glycemic control and a duration of T2D of app. 3.5 years before study inclusion. Consistent with our study, Artz et al. [24] measured hepcidin with a competitive assay and baseline hepcidin was comparable to ours (each app. 20 ng/mL) [24]. However, Artz et al. [24] included aging men with anemia, divided the study cohort according to the type of anemia, and found that TRT suppressed hepcidin levels in men with unexplained anemia, whereas hepcidin levels were unchanged in men with iron deficiency [24]. The authors concluded that TRT stimulated erythropoiesis associated with increased iron mobilization, but this effect was attenuated by iron deficiency [24]. Anemia is an important driver of hepcidin [30], but notably, all men in our study had normal hemoglobin and iron levels at study inclusion. Iron as well as the FGF23-phosphate-klotho pathway and calprotectin were unchanged during TRT, and delta hepcidin was not associated with changes in iron in the present study. These findings support that decreased hepcidin during TRT was not mediated by increased iron mobilization, in opposition to the conclusion by Artz et al. [24], at least after 24 weeks of TRT. Iron levels were unchanged during 24 weeks of TRT compared to placebo and therefore it is implausible that decreased hepcidin levels are linked to a higher consumption of iron as part of an increased erythropoiesis leading to the increased levels of hematocrit and hemoglobin.

In contrast to our study, Dhindsa et al. [17] included men with T2D of longer duration (app. 10 years) usingvarying antidiabetic treatment with the allowance to change anti-hyperglycemic medications during the trial, including insulin (50%), and supra-physiological testosterone dose was applied [17]. The study by Bachman et al. [25] included older men (>65 years) with mobility limitations and a high burden of chronic diseases, including 50% with heart disease, and the study was terminated due to increased CVD events in the TRT group [25]. Hepcidin production is stimulated by inflammation [20] and hepcidin levels are associated with obesity and the duration and severity of T2D [23,31]. In accordance, our level of hepcidin (20 ng/mL) was considerably lower compared to hepcidin levels in the study by Dhindsa et al. [17] and Bachman et al. [25], i.e., 200 ng/mL [17] and 100 ng/mL [25], respectively, which could suggest more inflammation [17,25]. Furthermore, Bachman et al. [25] and Dhindsa et al. [17] used conventional ELISA [17,25], and technical differences in the methods applied could affect the study results [32].

Insulin sensitivity, evaluated with a euglycemic clamp, as reported previously [10], was unchanged in the present study, and changes in hepcidin were not related to changes in body composition. These findings could suggest a direct effect of TRT on the synthesis of hepcidin from the liver, but this is only speculative. Our previous observation of reduced phosphate and calprotectin in aging men without T2D during TRT compared to placebo [29] was not observed in men with T2D.

Most of our patients were obese and they all fulfilled the criteria for having metabolic syndrome, which is the case for the majority of patients with T2D [33]. Obesity is a condition with extensive inflammation, closely related to liver steatosis [34], which could have affected our markers of inflammation since the liver is the main site for iron metabolism alterations and the main site for testosterone deactivation [35]. However, we observed unchanged ALAT and no change in hepatic fat content measured by MRS during TRT, as reported previously [12].

Strengths and limitations apply in the present study. Our study was strengthened by the inclusion of patients based on biochemical hypogonadism, testosterone evaluation by the gold standard method, i.e., liquid chromatography tandem mass spectrometry, per-protocol-analyses, a low dropout rate, and no dropouts due to adverse effects of the testosterone gel. The rates of absorption of the testosterone gel are relatively unreliable, and thus periodic testing of circulating testosterone is needed (Table 2). However, the effectiveness of our method with the administration of gel during 24 weeks has been proved by the significant increase in lean body mass and decrease in fat mass in the testosterone group compared to placebo [10]. In our study, male hypogonadism was treated through the continuous use of TRT for 24 weeks, which might not favor a possible ‘recovery’ of hypogonadism. It is probably not conceivable in a clinical RCT to apply other methods of treatment that would favor the normalization of testosterone. Furthermore, the placebo group had unchanged testosterone levels during 24 weeks and therefore we find it unlikely that the TRT group would regain normal testosterone levels despite intervals of a much shorter time of TRT followed by no TRT time.

The original study was primarily designed to evaluate the effects of TRT on lean body mass [10]. We acknowledge that the present study could be underpowered to identify significant alterations in some of the inflammatory markers.

In conclusion, TRT decreased hepcidin levels in older men with T2D and low bioavailable testosterone levels. Our data supported a direct anti-inflammatory effect of TRT after 24 weeks, as changes in hepcidin were not associated with changes in iron, lean body mass, or fat deposits.

## Figures and Tables

**Table 1 biomedicines-11-03184-t001:** Anthropometric and biochemical parameters in patients between groups.

	Testosterone Therapy (TRT)				Placebo				
	*n*	Baseline	24 Weeks	∆	*n*	Baseline	24 Weeks	∆	*p*-Value
Age (years) *	20	61.6 ± 5.7			19	59.4 ± 6.6			
Duration of T2D *	20	3.0 ± 2.2			19	3.9 ± 2.8			
BMI (kg/m^2^) *	20	30.6 (28.9−32.3)	30.7 (29.0−32.4)	0.1 (−0.2; 0.7)	19	30.8 (28.9−32.6)	30.7 (28.8−32.5)	−0.1 (−0.5; 0.6)	0.46
Luteinizing hormone (IU/L) *	20	4.2 (3.3–5.4)			19	3.1 (2.6−3.8)			
Hepcidin (µg/L)	20	20.6 ± 11.6	8.1 ± 6.1	−9.3 (−18.8; −5.2)	19	24.4 ± 11.4	18.6 ± 7.9	−5.0 (−11.4; 2.6)	**<0.01**
Iron (µmol/L)	20	11.1 (8.7; 13.1)	10.7 (9.6; 14.0)	0.9 (−3.1; 3.4)	19	11.1 (9.1; 12.1)	9.9 (8.5; 14.2)	−0.9 (−2.4; 1.9)	0.79
FGF23 (pmol/L)	20	0.9 (0.6; 1.3)	1.3 (0.9; 1.8)	0.2 (0.0; 0.6)	18	0.9 (0.5; 1.3)	0.9 (0.5; 1.4)	0.1 (−0.3; 0.3)	0.14
Phosphate (mmol/L)	20	1.1 (1.0; 1.2)	1.1 (1.0; 1.2)	−0.0 (−0.1; 0.1)	19	1.1 (1.0; 1.2)	1.1 (1.0; 1.2)	0.0 (−0.1; 0.1)	0.10
Klotho (ng/L)	20	439.5 (326.0; 529.5)	418.5 (354.0; 503.5)	−1.0 (−42.0; 19.5)	19	409.0 (315.0; 448.0)	399.0 (364.0; 558.0)	3.0 (−29.0; 74.0)	0.10
Calprotectin (µg/L)	20	22.0 (11.1; 29.3)	26.0 (16.4; 35.8)	5.8 (−3.4; 12.7)	19	25.1 (12.2; 32.2)	27.3 (16.7; 43.2)	2.3 (−4.7; 9.7)	0.78
Lean body mass (kg) *	20	61.9 ± 8.9	63.6 ± 8.4	1.9 (0.9; 2.6)	19	61.7 ± 7.5	61.5 ± 8.0	−0.2 (−0.8; 0.5)	**<0.01**
Total fat mass (kg) *	20	28.4 (24.7−32.6)	27.1 (23.3−31.6)	−1.2 (−1.9; 0.1)	19	27.1 (24.1−30.6)	27.2 (23.9−30.9)	0.1 (−1.0; 1.2)	**<0.01**
VAT/TAT (%) *	12	22.0 (18.4–26.3)	21.4 (17.1–26.6)	−0.1 (−2.3; 1.5)	13	23.2 (20.8–25.8)	22.6 (20.3–25.1)	−0.4 (−1.4; 0.2)	0.91
SAT/TAT (%) *	12	32.1 (28.0–36.7)	28.7 (24.7–33.5)	−2.6 (−4.9; −1.2)	13	31.4 (28.7–34.5)	31.2 (28.7–33.8)	−0.8 (−1.9; 1.0)	**<0.01**
TFA/TTA (%) *	12	30.7 (26.1–36.1)	27.3 (22.8–32.7)	−3.7 (−4.9; −2.4)	13	27.9 (24.0–32.3)	28.0 (24.2–32.4)	−0.8 (−1.7; −1.2)	**<0.01**
Hepatic fat/water ratio *	11	0.4 (0.1; 0 0.6)	0.3 (0.1; 0.7)	0.0 (−0.1; 0.0)	13	0.4 (0.2; 0.7)	0.2 (0.2; 0.5)	−0.10 (−0.3; 0.0)	0.12
Rd clamp (mg/min/kg fat-free mass) *	20	5.9 (5.1−6.8)	6.2 (5.3−7.2)	0.4 (−0.8; 1.2)	19	6.0 (5.3−6.8)	5.8 (5.0−6.6)	0.08 (−1.0; 0.5)	0.29
HbA1c (mmol/mol) *	20	47 (45−50)	50 (46−53)	2 (1; 4)	19	48 (44*–*51)	48 (44–52)	0 (−4; 4)	0.13
Haemoglobin, mmol/L *	20	9.0 (8.7–9.2)	9.3 (9.1–9.5)	0.4 (−0.1; 0.8)	19	9.1 (8.9–9.3)	8.9 (8.6–9.1)	−0.2 (−0.5; 0.0)	**<0.01**
Haematocrit, % *	20	43.1 ± 0.02	45.4 ± 0.02	2.5 (−0.5; 4.0)	19	43.3 ± 0.02	42.7 ± 0.02	−1.0 (−2.0; 1.0)	**<0.01**
HDL cholesterol (mmol/L) *	20	1.0 (0.9–1.1)	1.0 (0.8–1.1)	−0.1 (−0.1; 0.0)	19	0.9 (0.9–1.0)	1.0 (0.9–1.1)	0.1 (0.0; 0.2)	**<0.01**
Adiponectin (mg/L) *	20	7.5 (6.1–9.3)	6.8 (5.5–8.4)	−0.7 (−1.2; −0.2)	19	6.2 (5.1–7.6)	6.1 (5.2–7.2)	−0.2 (−0.8; 0.4)	**<0.05**
Leptin (µg/L) *	20	13.2 (10.0–17.5)	9.5 (6.8–13.2)	−3.7 (−5.4; −1.3)	19	11.6 (9.0–14.9)	12.4 (9.6–16.0)	0.4 (−1.8; 3.6)	**<0.01**

Data presented as geometric mean (95% CI), arithmetic mean ± standard deviation, or median (interquartile range), as appropriate. All delta (∆) values are presented as median (interquartile range). *p*-value refers to the placebo-controlled mean effect of interventions between groups (β) or non-parametric Wilcoxon rank-sum tests for ∆ values between groups, as appropriate. VAT, visceral adipose tissue. TAT, total abdominal tissue. SAT, subcutaneous adipose abdominal tissue. TFA, total fat area thigh. TTA, total thigh area. Previously published data (*).

**Table 2 biomedicines-11-03184-t002:** Hormone levels.

	Testosterone (*n* = 20)	Placebo (*n* = 19)	*p*-Value
Total testosterone (nmol/L) Baseline 3 weeks 24 weeks ∆24 weeks-baseline	7.1 (6.6; 11.9) 11.9 (9.8; 16.7) 22.1 (8.2; 34.3) 13.1 (−1.6; 23.3)	9.4 (8.1; 12.5) 9.8 (8.8; 11.7) 10.2 (8.4; 12.1) 1.4 (−0.02; 2.2)	0.056
Bioavailable testosterone (nmol/L) Baseline 3 weeks 24 weeks ∆24 weeks-baseline	4.0 (2.9; 4.9) 5.8 (3.9; 6.9) 10.2 (3.7; 23.9) 6.8 (−0.5; 18.2)	4.7 (4.0; 5.2) 4.9 (4.4; 5.3) 5.7 (4.6; 6.1) 0.8 (−0.1; 1.5)	0.046
Free testosterone (nmol/L) Baseline 3 weeks 24 weeks ∆24 weeks-baseline	0.20 (0.15; 0.26) 0.30 (0.24; 0.40) 0.54 (0.19; 1.11) 0.36 (−0.02; 0.85)	0.24 (0.21; 0.28) 0.25 (0.24; 0.27) 0.29 (0.23; 0.31) 0.04 (−0.00; 0.07)	0.046
SHBG (nmol/L) Baseline 3 weeks 24 weeks ∆24 weeks-baseline	32 (28–37) 30 (25–35) 27 (23–32) −5 (−10; −3)	29 (24–35) 27 (22–34) 29 (23–36) −3 (−6; 2)	0.03
17β-Estradiol (pmol/L) Baseline 24 weeks ∆24 weeks-baseline	12 (0; 48) 84 (36; 135) 43 (15; 97)	34 (0; 55) 35 (26; 52) 8 (−22; 29)	0.003
Dihydrotestosterone (nmol/L) Baseline 24 weeks ∆24 weeks-baseline	0.61 (0.47–0.80) 3.01 (1.95–4.65) 2.67 (0.71; 4.76)	0.59 (0.44–0.78) 0.59 (0.44–0.78) −0.01 (−0.14; 0.20)	<0.001

Data presented as geometric mean (95% CI) or median (interquartile range), as appropriate. All delta (∆) values are presented as median (interquartile range). *p*-value refers to the placebo-controlled mean effect of intervention between groups (β) or a non-parametric Wilcoxon rank-sum test for ∆ values between groups, as appropriate. All data have been reported previously, except for results after 3 weeks of TRT.

## Data Availability

The data presented in this study are available on request from the corresponding author.

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
