# Peer review of "Hepcidin Reduction during Testosterone Therapy in Men with Type 2 Diabetes: A Randomized, Double-Blinded, Placebo-Controlled Study"

_biomedicines, 2023, doi:10.3390/biomedicines11123184_

Round 1

Reviewer 1 Report

Comments and Suggestions for Authors

Please, read the attached file: Rev-Bim-2680615-23o-docx

Comments on the Quality of English Language

Please, check for the presence of awkward sentences (few), and revise the syntax overall.

Author Response

Thank you so much for taking the time to evaluate our manuscript so carefully. Please find our point-by-point response in the file attached. 

Reviewer 2 Report

Comments and Suggestions for Authors

The report presents a secondary analysis of data from a clinical trial investigating the effect of testosterone on metabolic status.  The focus is on hepcidin, a key regulator of iron loading in the body.  In agreement with other studies the main finding is that testosterone treatment decreases levels of hepcidin, compared with controls.  In the treatment group levels of hepcidin are reduced by more than 50%.  One would expect to see some changes (increases) in iron levels and / or increases in hemoglobin or hematocrit.

The authors focus on iron levels in their analysis, which don't change.  They note the increase in hematocrit [lines 166] but don't discuss the possibility that this can be linked to decreased hepcidin and hence higher absorption of iron.

Some of the data in the table are not clear - the delta values for hematocrit are too low. It should be about 2.2 (given as 0.03) and -0.6 (given as -0.01) for TRT and placebo groups respectively. Values for hepcidin, leptin and a few others appear inconsistent.

The authors rightly try to make the most of the data from their clinical trial but the discussion and analysis needs re-writing.  Reference to hepcidin as an cardiovascular risk marker [line12-13, 64-65] is too misleading.  There is no mention of increased hematocrit in the abstract.

Lines 49-50  "hepcidin increases as an acute phase reactant" - this phrase is unclear and could suggest hepcidin is an inflammatory marker, which it is not - levels are modulated by inflammatory conditions ie. inflammation can lead to increased expression from the liver and decreased iron in the circulation. The authors understand this; some clarification would improve the narrative.

Lines 189-190 and line 228 need clarification or explanation.  The suggestion is that testosterone directly affects hepcidin synthesis from the liver? By what mechanism?

Lines 191-192 "significant knowledge" seems an exaggeration; the study confirms observations of others concerning he effect of TRT on hepcidin. 

Lines 219-225: the discussion of the low values of hepcidin concentrations found in the study, about 20ng/mL, is useful and highlights the possibility that inflammation was not prevalent, which could be discussed.

Comments on the Quality of English Language

The English is generally very clear.  Some phrases that need clarification, described above.  Some inconsistent spelling: heamatocrit / hematocrit.

Author Response

Thank you for taking the time to revise our manuscript so carefully. Please find our reponse as an attached file.

Reviewer 3 Report

Comments and Suggestions for Authors

Hepcidin reduction during testosterone therapy in men with 2

type 2 diabetes. A randomized, double-blinded, placebo-con- 3

trolled study.

The study on the modulation of the hepcidin level by exogenous testosterone is important due to its anti-inflammatory effect and maintaining the serum iron level. Intoduction provides sufficient background and includes up-dated references. The research design is appropriated. Methods are adequately described. Table on anthropometric and biochemical parameters in patients between groups is systematized. Reults are clearly presented and conclusions are supported by the results.  

Author Response

Reviewer 3

Comments and Suggestions for Authors

The study on the modulation of the hepcidin level by exogenous testosterone is important due to its anti-inflammatory effect and maintaining the serum iron level. Intoduction provides sufficient background and includes up-dated references. The research design is appropriated. Methods are adequately described. Table on anthropometric and biochemical parameters in patients between groups is systematized. Reults are clearly presented and conclusions are supported by the results.  

Thank you so much for your kind review.

Round 2

Reviewer 1 Report

Comments and Suggestions for Authors

Please, read the attached text

Author Response

Dear Reviewer,

We are grateful for the constructive criticism helping us to improve our manuscript.

Please read our response in the document attached.

Kind regards and on behalf of the study group,

Line Velling Magnussen, MD
